# Advances in the Management of Central Nervous System Metastases from Breast Cancer

**DOI:** 10.3390/ijms232012525

**Published:** 2022-10-19

**Authors:** Jorge Avila, José Pablo Leone

**Affiliations:** 1Department of Internal Medicine, St Elizabeth’s Medical Center, 736 Cambridge St., Boston, MA 02135, USA; 2Department of Medicine, Tufts University School of Medicine, Boston, MA 02111, USA; 3Department of Medical Oncology, Dana-Farber Cancer Institute, Harvard Medical School, 450 Brookline Ave., Boston, MA 02215, USA

**Keywords:** breast cancer, brain metastases, metastatic breast cancer, central nervous system metastases

## Abstract

Central nervous system (CNS) metastases are common in breast cancer (BC) patients and are particularly relevant as new treatments for BC are prolonging survival. Here, we review advances in the treatment of CNS metastases from BC, including radiotherapy, systemic therapies, and the evolving role of immunotherapy. The use of radiotherapy and chemotherapy is the cornerstone of treatment for CNS metastases. However, new targeted therapies have recently been developed, including anti-HER2 agents and antibody–drug conjugates that have presented promising results for the treatment of these patients.

## 1. Introduction

Breast cancer (BC) is the second most common cancer overall and the most common among women, representing one of the leading causes of mortality among women [1]. Around 30% of patients with BC will develop brain metastases (BM) during the course of their disease, which can have a devastating effect on prognosis, functional status, and quality of life [2]. The incidence of BM seems to have increased in recent years, most likely due to the prolonged survival of patients, the development of more efficient treatments, and the availability of better imaging techniques that lead to the increased detection of this complication [3].

The risk of CNS metastases is higher in BC patients that are hormone receptor (HR)-negative or human epidermal growth factor receptor 2 (HER2)-positive or have a high tumor grade [4]. According to this, BC subtypes have different tendencies to metastasize to the brain. Studies have shown that 25–46% of triple-negative breast cancer (TNBC), 11–48% of HER-2 positive, 8–15% of luminal A, and 11% of luminal B BC patients can develop CNS metastases [5,6].

The median overall survival (MOS) after the development of brain metastases has been described as 9.3 months for the luminal subtype, 16.5 months for the luminal HER-2 subtype, 11.5 months HER2 subtype, and 4.9 months for the TN subtype [7].

Different indexes have been created to assess the prognostic factors of patients with breast cancer brain metastases (BCBM). The recursive partitioning assessment (RPA) divides patients into three groups, class I, II, and III, with median survival (MS) times of 7.7 months, 4.5 months, and 2.3 months, respectively [8]. The graded prognostic assessment (GPA) divides patients into four groups based on a score obtained from clinical and biological characteristics: from 0 to 1, with an MS of 2.6 months; from 1.5 to 2.5 with an MS of 3.8 months; 3 with an MS of 6.9 months; and from 3.5 to 4 with an MS of 11 months [9]. More recently, a breast-specific GPA was developed, which provided a more accurate description of survival for these patients. The MS values for GPA 0–1, 1.5–2, 2.5–3, and 3.5–4 were 6, 13, 24, and 36 months, respectively [10].

It has been shown that patients with HER2-positive BC have a significantly higher incidence of central nervous system (CNS) metastases after treatment with trastuzumab, probably secondary to improvements in systemic disease control and longer MOS associated with this pharmacological therapy [11]. The registHER study, a prospective observational study of over 1000 HER2-positive metastatic BC patients, showed that the MOS after the diagnosis of CNS metastases was improved from 3.8 to 17.5 months with the administration of trastuzumab [12].

One of the first treatments for BM was whole-brain radiation therapy (WBRT). Other important local therapy options available today include stereotactic radiosurgery (SRS) and neurosurgical resection [13]. Surgical resection followed by radiotherapy (RT) could be curative for a small, solitary BM [14]. However, the current treatment options for patients with extracranial disease and/or multiple BM remain mainly palliative [6]. Most recently, targeted systemic therapies and immunotherapy appeared in the multidisciplinary management of BM, leading to an improvement in intracranial control, survival, and neurocognitive preservation among these patients [13].

The blood–brain barrier (BBB), a neurovascular unit composed of endothelial cells, astrocytes, and pericytes [15], makes the treatment of BM challenging, as it forms a selective barrier between the CNS and the systemic circulation [16]. The BBB is disrupted during tumor progression and forms the blood–tumor barrier (BTB), which is more permeable than the BBB, allowing multiple drugs into the CNS to target the tumor and healthy brain parenchyma [17]. The permeability of the BTB varies depending on the subtype of cancer [18]. For example, BM from TNBC or basal-type BC may often disrupt the BBB, whereas BM from HER2-positive BC tend to preserve the BBB [19].

Although there is no doubt the BTB is more permeable than the BBB, it still significantly restricts the delivery of anticancer drugs and obstructs the systemic chemotherapeutics of brain tumors [20]. RT, besides providing cytotoxicity, can cause disruption of the BBB, resulting in increased permeability into the surrounding brain parenchyma [21]. A better understanding of both the BBB and BTB is needed to develop treatments with higher penetration to the CNS or increased manipulation of the CNS barrier.

In this review, we summarize local and systemic treatments for BCBM. First, we discuss in detail the existing data supporting the different treatment options, including surgical resection, various types of RT, chemotherapy, and targeted therapies. Then, we put all the data into context in a clinical algorithm for the recommended management of these patients. This may provide a better understanding of this topic and encourage the development of new strategies for the management of BCBM.

## 2. Local Treatment for BM

Local treatment for BM includes surgery, WBRT, SRS, fractionated stereotactic radiotherapy (fSRT), or a combination of these [6,22]. The decision of choosing one over another may be based on the estimated prognosis and the goals of the treatment [23]. Surgical resection is the standard treatment for a single BM, especially when it is large in size and causes a mass effect or obstructive hydrocephalus [24]. In patients with limited BM (defined as one to four BM), surgical resection and SRS are considered acceptable treatments as well. WBRT remains beneficial in certain situations and is often used in patients with widely disseminated BM [25].

The use of surgery is most often reserved for patients with a good performance status and good extracranial disease control (or absence of extracranial disease) [14]. However, surgery alone is considered inadequate for local control when compared to surgery plus RT [26] Surgery followed by RT improves MOS and symptom control compared with RT alone [27]. Studies have shown that, in patients with a single BM, WBRT after surgical resection reduces the rate of recurrence at the initial metastatic site and other brain sites [26].

The use of WBRT alone is indicated only in patients with more than ten BM for whom local treatment is not appropriate and in patients with new lesions on which additional SRS cannot be performed [28]. Studies have shown improvements in symptoms in 64–83% of patients after treatment with WBRT alone [29,30,31] and have also shown an increase in MOS from 1 month with no treatment to 3.7 months after WBRT [32]. However, this treatment is associated with toxicities, including dermatitis, alopecia, nausea, cerebral edema, and even cognitive deterioration [25], making the use of other techniques preferable over WBRT.

Patients with limited intracranial disease can be offered SRS, which delivers large doses of radiation to a well-defined area [33]. Recent studies affirmed that, while WBRT reduces the risk of local and regional intracranial disease progression relative to SRS alone, these benefits come at the cost of increased neurotoxicity without improved survival [34]. Kayama et al. evaluated whether salvage SRS alone within 21 days of surgery is as effective as postoperative WBRT on the OS of patients with 1–4 BM. The authors concluded that salvage SRS can be a standard therapy for patients with this number of BM, as it was observed to be noninferior to WBRT, with an MOS of 15.6 months in both arms (hazard ratio: 1.05; *p* for noninferiority = 0.027) [35].

Andrews et al. compared survival between WBRT alone and WBRT followed by SRS in patients with 1–3 BM and discovered that MOS did not differ between these two groups, but WBRT plus SRS resulted in better survival for patients with a single unresectable BM when compared to WBRT alone (MOS: 6.5 vs. 4.9 months, respectively; *p* = 0.039) [36].

Other studies have investigated the difference between SRS alone and WBRT plus SRS and showed that patients with 1-4 BM treated with the latter option did not have a longer survival (8 vs. 7.5 months; *p* = 0.42) [37]. This was also reported in patients with 1–3 BM by Brown et al. (*p* = 0.92) [34], who showed that using SRS alone could be a better treatment strategy for patients with less than five BM.

A multi-institutional trial of WBRT vs. SRS for patients with 1–4 BM after the resection of 1 metastasis showed that WBRT was associated with improved local control (80.6% vs. 60.5%; *p* = <0.001) and intracranial control (78.6% vs. 54.7%, *p* < 0.001). However, it did not show any differences in OS, and cognitive deterioration was higher at 6 months in the WBRT group [38].

In patients with five or more BM, it is still unclear whether WBRT or SRS should be used. Li et al. compared SRS with WBRT for patients with 4–15 BM, and they described no difference in MOS between the two groups (10.4 vs. 8.4 months; *p* = 0.45); however, in the same study, SRS reduced the risk of cognitive deterioration; suggesting that there may be a benefit in avoiding WBRT in this setting [39].

As mentioned before, fSRT has also been evaluated for BCBM. fSRT is less costly and more comfortable for patients [40], and retrospective reviews comparing fSRT to SRS have shown no difference in the local control of BCBM [41]. Meanwhile, emerging studies suggest improved local control in fSRT when compared to SRS for BM [42,43]. However, there is limited information on fSRT when compared to WBRT. The use of fSRT in 1–10 BM is currently being evaluated in the NCT04061408 study [44].

Hippocampal-avoidance WBRT (HA-WBRT) has been shown to be effective in protecting against cognitive decline when compared to standard WBRT [45]. This technique is considered safe, as BCBM has a low risk of metastases in the hippocampal region [46].

Table 1 summarizes the major clinical trials for BCBM treated with RT.

## 3. Systemic Therapies for BCBM

Plasma proteins such as immunoglobulin G (IgG) molecules generally do not cross the brain capillary endothelial wall, which forms the BBB under normal conditions. However, certain monoclonal antibodies in the circulation cross the BBB by a process of receptor-mediated transcytosis [47]. In this case, the monoclonal antibody is directed against a specific receptor located on the luminal membrane of the brain capillary endothelium, and the monoclonal antibody binds to exofacial epitopes on the BBB receptors. The reverse transcytosis of IgG molecules across the BBB is most likely mediated by an Fc receptor situated on the abluminal membrane of the brain capillary endothelium, and rat models have demonstrated a rapid IgG efflux with a half-life of 48 min [48]. In the following paragraphs, we will mention different drugs that have been evaluated for BCBM. However, their passage through the BBB/BTB is still under study.

### 3.1. Human Epidermal Growth Factor Receptor 2 (HER2)

#### 3.1.1. Trastuzumab

Trastuzumab is a monoclonal antibody directed against the extracellular domain of HER2. However, it has a large molecular weight that makes it difficult to cross the BBB, and regular intravenous (IV) administration may not be effective for BCBM [49].

Under impaired BBB conditions such as meningeal carcinomatosis or RT, trastuzumab levels in cerebrospinal fluid (CSF) increase; this evidence supports the concept of continuing trastuzumab therapy in patients with BM treated by RT. Monitoring trastuzumab in the serum and CSF may enable individualized therapies in BCBM [50].

Nonetheless, Park et al. demonstrated that patients receiving trastuzumab for BCBM had a significant longer time to death (14.5 vs. 4.0 months; *p* = 0.0005) [51], and other authors discovered that this treatment appeared to prolong the MOS by controlling extracranial disease [52,53]. The registHER study also revealed that treatment with trastuzumab after the diagnosis of BCBM significantly decreased the risk of death [12]. Dawood et al. showed that patients with HER2-positive disease treated with trastuzumab had a longer median time to CNS metastasis compared with similar patients who never received trastuzumab (13.1 vs. 2.1 months; *p* = 0.0008) [54].

Several case reports have described a benefit of intrathecal trastuzumab administration to treat carcinomatous meningitis in patients with HER2-overexpressing metastatic BC [55,56]. Bousquet et al. described the effect of intrathecal trastuzumab in a patient with HER-2 BCBM, and after maintaining an efficacious concentration of this drug in the CSF, they were able to achieve the stabilization of brain and epidural metastases previously resistant to radiation and chemotherapy [57]. More recently, Kumthekar et al. evaluated intrathecal trastuzumab in a multicenter phase I/II study on patients with HER2-positive leptomeningeal disease (LMD). Partial responses were observed in 19.2% of the patients, and stable disease was seen in 50% of the participants. This study suggests the potential for improved outcomes in HER2-positive LMD [58]. However, further analysis needs to be conducted for the evaluation of intrathecal trastuzumab.

Trastuzumab was the first established therapy targeted against HER2-positive BCBM; unfortunately, despite initial success, patients developed disease progression [18].

#### 3.1.2. Trastuzumab Emtansine (T-DM1)

Trastuzumab emtansine (T-DM1) is an antibody–drug conjugate composed of trastuzumab linked to the cytotoxic agent DM1 (a maytansine derivative), and it has been evaluated in multiple studies of patients with HER-2 positive BCBM, with reports of benefits in the median progression-free survival (PFS) [59,60].

EMILIA, a phase III trial, compared the efficacy of T-DM1 alone vs. lapatinib + capecitabine in HER2-positive advanced BC previously treated with trastuzumab and a taxane, and it showed that the T-DM1 group had improvements in PFS and OS (*p* < 0.001). In patients with treated asymptomatic CNS metastases at baseline, OS was improved in the T-DM1 group when compared to the lapatinib + capecitabine group (MOS: 26.8 vs. 12.9 months; *p* = 0.008) [61].

In 2020, Montemurro et al. published data from a subgroup of 398 patients with BM from the KAMILLA trial, a single-arm phase IIIb study of T-DM1 in patients with HER2-positive locally advanced/metastatic BC. The study confirmed the efficacy and safety of this drug, with a BM response rate of 21.4%, a median PFS of 5.5 months, and an MOS of 18.9 months [62].

#### 3.1.3. Trastuzumab Deruxtecan (T-DXd)

Trastuzumab Deruxtecan (T-DXd) is an antibody–drug conjugate of trastuzumab and an exatecan derivative (topoisomerase 1 inhibitor). The DESTINY-Breast01 trial investigated the efficacy of this drug in HER2-positive metastatic breast cancer (MBC) previously treated with T-DM1 and showed a response rate of 60.9% and a median PFS of 16.4 months [63].

DESTINY-Breast03, a phase III trial that compared T-DXd vs. T-DM1 in HER2 + MBC patients, showed that the estimated 12-month OS was 75.8% for the T-DXd group vs. 34.1% for the T-DM1 group (*p* = 7.8 × 10^−22^) [64]. Approximately 15% of the patients in each arm had the presence of brain metastases at baseline; among these patients, the median PFS was 15 months for T-DXd vs. 3 months for T-DM1. In addition, intracranial responses were also higher in the T-DXd arm. Complete responses were observed in 27.8% of T-DXd patients and 2.8% of T-DM1 patients. Partial responses were observed in 36.1% and 30.6% of T-DXd and T-DM1 patients, respectively [65].

DESTINY-Breast 04, a phase III trial that included patients with BM, compared T-DXd vs. the treatment of physician’s choice (TPC) in previously treated HER2-low advanced breast cancer and showed that patients treated with T-DXd presented significantly longer PFS and OS than the TPC group (23.9 vs. 17.5 months; *p* = 0.003) [66].

#### 3.1.4. Pertuzumab

Pertuzumab is a humanized monoclonal antibody that inhibits the dimerization of HER2 with other HER receptors and can be considered for the treatment of CNS metastases [67]. The phase III trial CLEOPATRA compared pertuzumab, trastuzumab, and docetaxel with placebo, trastuzumab, and docetaxel; the results from this trial showed that the median time to progression in the CNS was 11.9 in the placebo group and 15.0 months in the pertuzumab group (*p* = 0.0049); however, the incidence of CNS metastases as the first site of disease progression was similar in both arms (13.7 % vs. 12.6%) [68]. The final results from this study stated that the MOS was improved in the group that received pertuzumab when compared to the placebo group (57.1 vs. 40.8 months, hazard ratio: 0.69) [69].

In the phase II PATRICIA study, patients with HER2-positive MBC with CNS metastases received pertuzumab plus high-dose trastuzumab, and although the overall CNS response rate was modest (11%), 68% of patients experienced a clinical benefit at 4 months [70].

#### 3.1.5. Lapatinib

Lapatinib is a dual tyrosine kinase inhibitor of the EGFR and HER2 that is able to cross the BBB due to its low molecular weight [17]. Studies have shown that single-agent lapatinib is active in patients with HER2-positive BCBM, with a modest response between and 2.6% and 6% [71,72]. In order to increase the response rate, lapatinib has been combined with capecitabine. Studies have shown that this combination is superior to capecitabine alone in HER2-positive MBC [73] as well as in BCMB, with an objective response rate (ORR) of 20%; in addition, volumetric reduction was observed in the CNS lesions of the patients treated with this combined regimen [72].

LANDSCAPE, a single-arm phase II trial, investigated the efficacy of combining lapatinib and capecitabine in patients with an initial recurrence of BM not previously treated with WBRT. This study presented a CNS response rate of 65.9% [74].

CEREBEL, a phase III randomized study designed to investigate the incidence of the CNS being the first progression in patients with HER2-positive MBC, compared lapatinib + capecitabine vs. trastuzumab + capecitabine, and it showed that trastuzumab + capecitabine had a longer PFS (hazard ratio: 1.30; *p* = 0.021) [75].

A pooled analysis of 12 studies about the use of lapatinib + capecitabine or lapatinib alone in HER2-positive BCBM demonstrated an ORR of 21.4%, a PFS of 4.1 months, and an OS of 11.2 months [76].

The use of lapatinib combined with trastuzumab has also been studied. Patients with HER2-positive MBC or recurrent breast cancer (RBC) with BM treated with this combination therapy had a significantly longer survival than those treated with trastuzumab alone, lapatinib alone, or no HER2-targeted therapy (*p* < 0.001) [77]. Other studies have shown that patients treated with lapatinib + taxane had a significantly shorter PFS (hazard ratio: 1.48; *p* < 0.001) than those treated with trastuzumab + taxane [78].

Lin et al. have also studied the combination of lapatinib and WBRT, which has shown a higher ORR (79%) when compared to the historical results of WBRT alone [79].

#### 3.1.6. Neratinib

Neratinib is an irreversible inhibitor of EGFR, HER1, HER2 and HER4 and has demonstrated activity both as a single agent and in combination with paclitaxel [80].

NEfERT-T, a phase II trial, compared the use of neratinib + paclitaxel vs. trastuzumab + paclitaxel in HER2-positive MBC and showed that the incidence of CNS metastases was lower (8.3% vs. 17.3%; *p* = 0.002), and the time to CNS metastases was delayed in the group receiving neratinib + paclitaxel; however, both groups had similar median PFS values [81].

The TBCRC 022 trial evaluated neratinib as a single agent in HER2-positive BCBM and presented a low CNS objective response of 8%. However, the trial was expanded and showed that neratinib + capecitabine resulted in a CNS response of 49% in the lapatinib-naïve group vs. 33% in the cohort of lapatinib pretreated patients, reaffirming the synergy observed between lapatinib and capecitabine [82].

The NALA trial compared neratinib + capecitabine vs. lapatinib + capecitabine in HER2-positive MBC and demonstrated a longer PFS in the neratinib group (8.8 vs. 6.6 months; *p* = 0.003). In this study, the cumulative incidences of intervention for CNS disease were 22.8% in the neratinib group and 29.2% in the lapatinib group (*p* = 0.043) [83].

#### 3.1.7. Tucatinib

Tucatinib is a specific inhibitor of HER2 and a substrate of *p* glycoprotein (*p*-gp), which would be expected to limit distribution to CNS. However, a reduced number of efflux transporters, acid interstitial pH, and leaky junctions enhance tucatinib permeability into the tumor [84].

Different studies have shown an increase in CNS response (12%) and better PFS when combined with trastuzumab, and when combined with T-DM1, the results have presented a median PFS of 6.7 months and a 36% CNS response [85,86].

The combination of tucatinib + trastuzumab + capecitabine has been explored in various studies, including a phase I trial that resulted in a CNS response of 42% and a median PFS of 6.7 months [87]. HER2CLIMB, a phase II trial that studied the efficacy if this triple combination on the intracranial response and survival in HER2-positive BCBM, showed a significant improvement in PFS, with a 1-year rate of 24.9% in the tucatinib group vs. 0 in the placebo group (*p* < 0.001) and median PFS values of 7.6 vs. 5.4 months, respectively [88]. A subgroup of patients from this trial was analyzed to determine the intracranial response (47.3% vs. 20%), CNS PFS (9.9 vs. 4.2 months), MOS (18.1 vs. 12.0 months), and a reduced risk of death (42%) in the tucatinib arm [89].

Data from HER2CLIMB also demonstrated a 62% reduction in disease progression or death and a longer PFS (7.6 vs. 4.1 months) in the patients receiving tucatinib vs. placebo [90,91].

The final OS analysis from HER2CLIMB showed that the MOS was 24.7 months for the tucatinib combination vs. 19.2 months for the placebo group (*p* = 0.004) [92].

Currently, HER2CLIMB-02 is evaluating the use of tucatinib + T-DM1 in patients with HER2-positive MBC.

#### 3.1.8. Pyrotinib

Pyrotinib, an irreversible pan-HER receptor inhibitor, has shown higher efficacy when compared to lapatinib in HER2-positive MBC [93]. However, its benefit in BCBM is still unknown. Nonetheless, there is an ongoing phase II trial of pyrotinib + vinorelbine in HER2-positive BCBM patients (NCT03933982).

PERMEATE, a phase II trial that studied pyrotinib + capecitabine for patients with HER2-positive BCBM, demonstrated a intracranial objective response rates of 74.6% in radiotherapy-naïve patients and 42.1% in those with progressive disease after RT [94].

### 3.2. Vascular Endothelial Growth Factor (VEGF)

#### Bevacizumab

Bevacizumab is a recombinant humanized monoclonal antibody against VEGF and has been studied in MBC. A phase III trial compared paclitaxel + bevacizumab vs. paclitaxel alone in MBC and observed a longer PFS in the combined treatment group; however, there was no improvement in OS, and unfortunately this study did not include CNS metastases [95].

A phase II trial evaluated the efficacy of carboplatin and bevacizumab in BCBM, and presented an ORR of 63%, a median PFS of 5.62 months, and an MOS of 14.10 months [96]. Another phase II study in patients with BCBM demonstrated a CNS ORR of 77% with bevacizumab followed by etoposide and cisplatin [97].

### 3.3. Phosphatidylinositol 3-kinase (PI3K)/Mammalian Target of Rapamycin (mTOR)

Approximately 40% of HR-positive, HER2-negative BCs display PIK3CA mutations.

Although certain studies have proven the efficacy of alpelisib in the treatment of MBC [98,99], these have excluded patients with active or untreated BCBM. Four cases of patients with BCBM have been described in the literature about reduced size or stable disease with the use of alpelisib in combination with hormone therapy [100].

Buparlisib, a pan-class I PI3K inhibitor, has been shown to penetrate the BBB [101] and presented a better PFS (hazard ratio: 0.67; *p* = 0.0003) in the phase III BELLE trial when comparing buparlisib + fulvestrant vs. placebo for MBC patients [102].

Everolimus was also analyzed by Hurvitz et al. in a phase Ib/II single-arm trial investigating a triple therapy of lapatinib + everolimus + capecitabine in patients with HER2-positive BCBM. The results showed a 27% CNS ORR at 12 weeks and 6.2 months of PFS [103]. Everolimus was also studied in combination with trastuzumab + vinorelbine in BCBM and resulted in an intracranial response rate of 4% and a CNS clinical benefit of 27% at 6 months [104]. Paxalisib, a dual PI3L/mTOR inhibitor, is being studied to evaluate its clinical efficacy on HER2-positive BCBM [18].

### 3.4. Cyclin-Dependent Kinase 4/6

CDK 4–6 inhibitors block the accelerated cell cycle transition from G1 to S phase and therefore suppress cell cycle dysregulation [67].

Abemaciclib, a CDK 4/6 inhibitor, has shown improvement in ORR and PFS in patients with HR-positive BC when used in combination with either fulvestrant or aromatase inhibitors [105]. A phase II trial of abemaciclib that included patients with leptomeningeal metastases as well as BM treated with surgical resection showed that abemaciclib achieved therapeutic concentrations in BM tissue, confirming that this drug crosses the BBB [106].

Another phase II trial of abemaciclib in HR-positive/HER2-negative BCBM revealed a 38% intracranial response and a intracranial benefit rate of 25% [107].

Ribociclib combined with letrozole, presented positive results in the CompLEEment-1 study, a phase IIIb trial including HR-positive/HER2-negative MBC patients, with an ORR of 20.5% [108], and patients from this study with CNS metastases presented favorable results, with an ORR of 41.2% [109].

The MONALEESA trials studied the use of ribociclib in combination with endocrine therapy (ET) in patients with MBC. MONALEESA-2 and MONALEESA-7 included patients with CNS metastases, and in both of them the addition of ribociclib with ET resulted in measurable tumor reductions conferring an advantage in ORR (54.5% vs. 38.8%; *p* < 0.001 and 51% vs. 36%; *p* < 0.001, respectively) [110].

Palbociclib is also being evaluated to demonstrate its efficacy in BCBM, and we await the results from ongoing trials.

### 3.5. Poly (ADP-Ribose) Polymerase Inhibitors

Veliparib + WBRT was studied in a phase I trial that included BCBM and an MOS of 7.7 months was reported, compared to 4.9 months based on historical data [111].

The phase III EMBRACA trial explored the efficacy of talazoparib treatment in patients with BRCA-mutated MBC. The median PFS was higher (8.6 vs. 5.6 months) and the ORR was also higher (62.6% vs. 27.2%; *p* < 0.001) in the talazoparib group, and a subanalysis of patients from this study with BM presented similar results [112].

OlympiAD, a phase III trial, evaluated olaparib for MBC, and while there was no statistically significant improvement in OS with olaparib, there was a possibly of meaningful OS benefit among patients who had not received chemotherapy for MBC [113].

In BROCADE3, a phase III trial that compared veliparib + carboplatin + paclitaxel vs. placebo + carboplatin + paclitaxel in BRCA-mutated advanced BC, the addition of veliparib to carboplatin + paclitaxel provided a longer PFS than that observed in the placebo group (14.5 vs. 12.6 months; *p* = 0.0016). However, patients with active brain metastases were excluded from this study, and only 5% of the participants had a history of BCBM [114].

The SWOG trial, a phase II study of cisplatin +/− veliparib in MBC, showed that the addition of veliparib significantly improved PFS (5.7 vs. 4.3 months; *p* = 0.023) [115].

### 3.6. Sacituzumab Govitecan

Sacituzumab govitecan (SG) is an antibody–drug conjugate composed of an antitrophoblast cell-surface antigen 2 (Trop-2) IgG1 kappa antibody coupled to SN-38 (the active metabolite of irinotecan) that has presented positive results when combined with single-agent TPC [116]. Currently the SWOG 2007 study (NCT04647916) is evaluating intracranial ORR in patients with HER2-negative BCBM.

The primary results from TROPiCS-02, a randomized phase III study of SG vs. TPC in patients with HR-positive/HER2-negative MBC, demonstrated that SG improved median PFS when compared to TPC (5.5 vs. 4 months, hazard ratio: 0.66; *p* = 0.0003) as well as PFS at 6 and 12 months (46% vs. 30% and 21% vs. 7%, respectively) [117].

### 3.7. Etirinotecan Pegol

Etirinotecan pegol (EP), a novel long-acting topoisomerase-1 inhibitor, was studied in patients with BCBM by Cortes et al. in the phase III BEACON trial that compared EP to TPC. EP was associated with a significant reduction in the risk of death when compared to TPC (*p* < 0.01). The MOS was also higher on the EP group (10 vs. 4.8 months) [118].

### 3.8. Immune Checkpoint Inhibitors for BCBM

The expression of the programmed cell death protein 1 receptor ligand (PD-L1) has been suggested to be a therapeutic target of immune checkpoint inhibitors in BCBM [119].

KEYNOTE-012, a phase Ib randomized trial of single-agent pembrolizumab that included patients with advanced PD-L1-positive TNBC (and other types of malignancies) with treated and stable BM, demonstrated an overall response rate of 18.5% [120].

KEYNOTE-355, a phase III trial that studied pembrolizumab + chemotherapy in advanced TNBC, including patients with previously treated BM, resulted in a significantly longer MOS in the pembrolizumab-chemotherapy group vs. the placebo-chemotherapy group (23 vs. 16.1 months, hazard ratio: 0.73; *p* = 0.0185) [121].

Impassion130, a phase III clinical trial investigating Atezolizumab + Nab-Paclitaxel vs. placebo + Nab-paclitaxel reported a prolonged MOS among patients with metastatic TNBC (including BCBM) in both the intention-to-treat population (21.3 vs. 17.6 months, hazard ratio: 0.84; *p* = 0.08) and the PD-L1 + subgroup (25 vs. 15.5 months, hazard ratio: 0.62; *p* < 0.001) [122].

New studies are being developed to investigate the efficacy of immune checkpoint inhibitors in BCBM, including a phase I/II trial evaluating pembrolizumab + SRS (NCT03449238), a phase I study analyzing SRS after nivolumab (NCT03807765), and a randomized phase III trial exploring the use of nivolumab and iplimumab in solid tumor BM (NCT04434560), among others [67].

### 3.9. Other Therapies for BCBM

There are currently studies recruiting patients to evaluate the efficacy of chimeric antigen receptor (CAR)-T cell therapies in HER2-positive BCBM (NCT02442297 and NCT03696030). Other trials investigating the effects of dendritic cell therapy for BCBM (NCT03638765) and proteome-based immunotherapy of BCBM (NCT01782274) are under development [123].

ANG1005, a novel taxane derivative consisting of three paclitaxel molecules covalently linked to Angiopep-2, designed to cross the BBB and BTB and to penetrate malignant cells via the lipoprotein-receptor-related protein 1 (LRP-1) transport system, has demonstrated intracranial response, symptom improvement, and prolonged overall survival compared to historical controls in different studies on BCBM [124,125]. Unlike native paclitaxel, ANG1005 was proven not to be a substrate for the multiple drug resistance (MDR) efflux pump in in vitro studies and has shown similar brain penetration in mice. This indicates that ANG1005 can be used to treat both peripheral metastatic BC and BCBM, even if the tumor develops resistance to conventional taxanes [126].

The efficacy of these therapies needs further evaluation.

Table 2 summarizes the majority of the systemic therapies for BCBM.

## 4. Clinical Approach

The treatment of BCBM will depend on the number of BM, the ER/PR and HER2 status of the tumor, and the control or progression of intra- and extracranial disease (Figure 1).

For patients with a single BM, surgical resection is suggested, as it can improve OS, especially in symptomatic patients when systemic disease is well-controlled. If patients present one to four BM, surgery + SRS with or without WBRT should be considered to improve local control [127]. Memantine and hippocampal avoidance should be offered to patients with no hippocampal lesions and 4 months or more of expected survival [128].

In patients with HER2-positive metastatic disease and limited asymptomatic intracranial disease, upfront systemic therapy can be the initial treatment instead of RT. For patients who are T-DM1-naïve, a deferral of RT and treatment with T-DM1 with the intent to treat BM may be appropriate [62].

For patients who have no evidence of extracranial disease and achieve an excellent clinical response rate after local treatment for BCBM, there are no prospective data to inform the benefits of systemic treatment.

Patients with progressive/new intracranial and progressive extracranial disease or no feasible local therapy option should undergo treatment with systemic therapy based on their HER2 status.

If patients have HER2-negative disease, single-agent chemotherapy with drugs such as fluorouracil, capecitabine, platinums, or doxorubicin has been described to have activity against CNS metastases [129,130]. Patients with hormone-receptor-positive disease can receive tamoxifen, aromatase inhibitors, or other similar agents; however, ET should not be used alone as a monotherapy, and local therapies are still necessary [131,132,133]. This is where CDK 4–6 inhibitors can be used as well.

If patients have HER2-positive disease, patients should receive a combination of Pertuzumab + Trastuzumab + Taxane-based chemotherapy. If patients progress after this, the use of a combination with Tucatinib + Trastuzumab + Capecitabine is recommended [86]. If patients progress after the previous treatment, the data recommend the use of T-DXd [134]. Meanwhile, some authors recommend the use of T-DM1 instead [61]. Additional lines may include the use of neratinib in combination with capecitabine, which has shown an intracranial response in BCBM [82].

## 5. Conclusions

The presence of CNS metastases in BC is associated with limited survival, and its incidence is increasing with the new development of BC therapies. However, although still limited, the treatment of BCBM has shown promising results for a better prognosis and prolonged PFS in multiple studies. Local treatments, including surgery, WBRT, and SRS, are becoming more conservative, limiting cognitive decline, and enhancing quality of life.

Targeted therapies for BCBM have been established, and multiple clinical trials with drugs directed against HER2, VEGF, mTOR, PI3K, EGFR, and CKD4/6; immune checkpoint inhibitors; and even CAR-T-cell therapy have presented positive outcomes in this disease.

The promising results from the trials summarized in this article should encourage further studies of the treatment of BCBM.

## Figures and Tables

**Figure 1 ijms-23-12525-f001:**
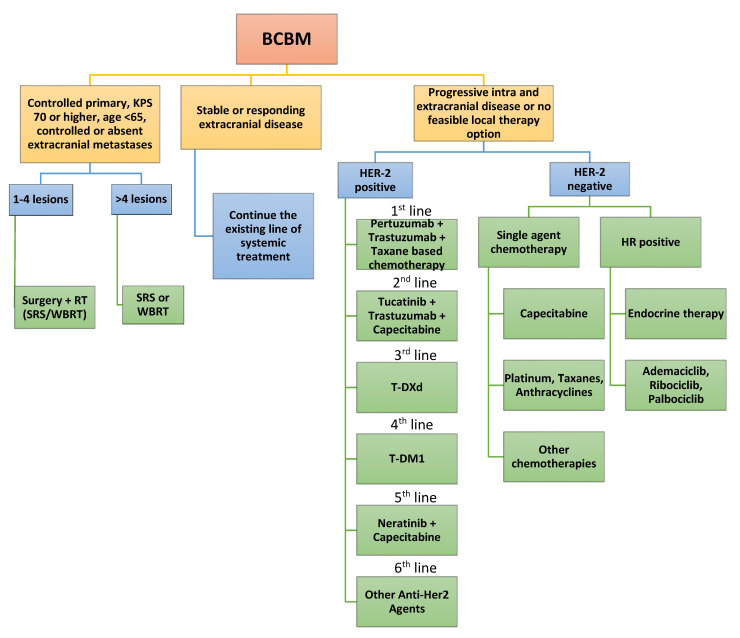
Suggested treatment algorithm for patients with breast cancer brain metastases. Abbreviations: BCBM, breast cancer brain metastases; KPS, Karnofsky Performance Scale; RT, radiotherapy; SRS, stereotactic radiosurgery; WBRT, whole-brain radiation therapy; HER-2, human epidermal growth factor receptor 2; T-DXd, trastuzumab-deruxtecan; T-DM1, trastuzumab-emtansine; HR, hormone receptor.

**Table 1 ijms-23-12525-t001:** Major clinical trials of local treatments for BCBM.

Treatment	Author	Clinical Trial Number	Study Population	Phase	Primary Outcome
Surgery plus WBRT vs. WBRT alone	Patchell et al.		Single BM	III	OS
Surgery plus WBRT vs. surgery alone	Patchell et al.		Single BM	III	Recurrence of tumor in the brain
Preoperative SRS vs. postoperative SRS	(Ongoing)	NCT03741673	BM under 4 cm for single fraction and over 7 cm for multifractional therapy	III	LMD-free rate
WBRT with memantine vs. WBRT without memantine	Brown et al.	NCT0056685	Pathologically proven BM	III	Cognitive function
Preoperative SRS	(Ongoing)	NCT03368625	1–6 BM (2–4 cm)	II	Radiation toxicity
Postoperative SRS vs. WBRT	Brown et al.	NCT01372774	Single BM	III	Cognitive-deterioration-free survival and OS
Salvage SRS vs. postoperative WBRT	Kayama et al.	JCOG0504	1–4 resected BM with only one lesion > 3 cm	III	OS
WBRT alone vs. WBRT followed by SRS	Andrews et al.	NCT00002708	1–3 BM	III	OS
SRS alone vs. SRS plus WBRT	Brown et al.	NCT00377156	1–3 BM	III	Cognitive deterioration at 3 months
HA-WBRT + memantine vs. WBRT + memantine	Brown et al.	NTG Oncology CC001 (NCT02360215)	BM outside a 5 mm margin around either hippocampus	III	Time to cognitive function failure
SRS vs. HA-WBRT + memantine	(Ongoing)	NCT03550391	5–15 BM	III	OS and neurocognitive PFS
SRS or surgery with/without WBRT	Kocher et al.	NCT00002899	1–3 BM	III	Time to PS deterioration more than 2
SRS plus WBRT vs. SRS alone	Aoyama et al.	C00000412	1–4 BM, each under 3 cm	III	OS
SRS for 2–4 BM vs. 5–10 BM	Yamamoto et al.	UMIN000001812	Patients with BM who received SRS	III	OS
SRS vs. WBRT	Lie et al.	NCT01592968	4–15 untreated nonmelanoma BM	III	Local control rate and proportion of patients with neurocognitive decline at 4 months
SRS vs. HA-WBRT	(Ongoing)	NCT0307507	5–20 BM	III	QoL
HA-WBRT	Gondi et al.	NCT01227954	BM outside a 5 mm margin around either hippocampus	II	Cognitive function
FSRT	(Ongoing)	NCT0406140	1–10 HER2-positive BCBM	II	Intracranial local tumor control rate

Abbreviations: BC, breast cancer; MBC, metastatic breast cancer; BM, brain metastases; BCBM, breast cancer brain metastases; WBRT, whole-brain radiation therapy; SRS, stereotactic radiosurgery; HA-WBRT, hippocampal-avoidance whole-brain radiotherapy; OS, overall survival; PFS, progress-free survival; QoL, quality of life; LMD, leptomeningeal disease; FRST, fractionated stereotactic radiotherapy.

**Table 2 ijms-23-12525-t002:** Major clinical trials of systemic treatments for BCBM.

Treatment	Author	Clinical Trial Name (Number)	Study Population	Phase	Primary Outcome
Intra-arterial cerebral infusion of trastuzumab	(Ongoing)	NCT02571530	HER2-positive BCBM	I	MTD, adverse events, and dose-limiting toxicities
Lapatinib	Lin et al.	EGF 105084 (NCT00263588)	Progressive HER2-positive BCBM after prior trastuzumab and cranial RT	II	ORR in CNS
Lapatinib + capecitabine vs. lapatinib + topotecan	Lin et al.	NCT00437073	HER2-positive BC with progressive BCBM after trastuzumab and cranial RT	III	CNS objective response
Lapatinib + capecitabine vs. trastuzumab + capecitabine	Pivot et al.	CEREBEL (NCT00820222)	HER2-positive MBC without BM	III	Incidence of CNS metastases as first site of relapse
Intermittent high-dose lapatinib + capecitabine	Morikawa et al.	(NCT02650752)	HER2-positive BCBM	I	MTD
Lapatinib + capecitabine	Bachelot et al.	LANDSCAPE (NCT00967031)	HER2-positive BCBM not previously treated with WBRT, capecitabine, or lapatinib	II	ORR in CNS
Lapatinib + capecitabine vs. capecitabine alone	Geyer et al.	EGF100151 (NCT00078572)	ABC with progression on trastuzumab	III	Time to progression
Lapatinib + everolimus + capecitabine	Hurvitz et al.	TRIO-US B-09 (NCT01783756)	HER2-positive BCBM	I/II	CNS ORR
Lapatinib + WBRT	Lin et al.	NCT00470847	HER2-positive BCBM	I	Maximum tolerated dose of concurrent lapatinib with WBRT
Lapatinib + WBRT	Christodoulou et al.	NCT01218529	BM from HER2-positive BC or lung cancer	II	Response rate in brain as assessed by volumetric analysis in brain MRI
Lapatinib + WBRT/SRS vs. WBRT/SRS	Kim et al.	NCT01622868	HER2-positive BCBM	II	CR rate in measurable BM
Pertuzumab + high-dose trastuzumab	(Ongoing)	PATRICIA (NCT02536339)	HER2-positive progressive BCBM after RT	II	Percentage of participants with ORR in the CNS
Pertuzumab + trastuzumab and docetaxel vs. trastuzumab + docetaxel	Swain et al.	CLEOPATRA (NCT00597190)	HER2-positive locally recurrent, unresectable, or MBC without prior chemotherapy or biologic therapy for their advanced disease	III	PFS
T-DM1	Montemurro et al.	KAMILLA (NCT01702571)	HER2-positive locally advanced or MBC with prior HER2-targeted therapy and chemotherapy	III	Best overall response rate and clinical benefit rate
T-DM1 vs. lapatinib	Krop et al.	EMILIA (NCT00829166)	HER2-positive ABC previously treated with prior HER2-targeted therapy and chemotherapy	III	Percentage of participants with progressive disease or death, PFS, and OS
T-DM1 alone vs. T-DM1 plus metronomic temozolomide	(Ongoing)	NCT03190967	HER2-positive BCBM treated with SRS	I/II	MTD of temozolomide when used with T-DM1
Neratinib	Chan et al.	ExteNET (NCT00878709)	Stage II and IIIC HER2-positive BC with node-positive disease	III	Invasive disease-free survival at year 2
Neratinib + paclitaxel vs. trastuzumab + paclitaxel	Awada et al.	NEfERT-T (NCT00915018)	Previously untreated recurrent and/or metastatic HER2-positive BC	III	PFS
Neratinib + capecitabine	Freedman et al.	TBCRC 022 (NCT01494662)	Measurable, progressive, HER2-positive BCBM	II	ORR
Neratinib + capecitabine vs. lapatinib + capecitabine	Saura et al.	NALA trial (NCT01808573)	HER2-positive MBC with two or more previous HER2-directed MBC treatments	III	PFS and OS
Afatinib vs. Afatinib + 2Gy RT vs. afatinib + 4Gy RT	(Ongoing)	NCT02768337	BM from BC or lung cancer	I/II	Ratio of afatinib concentration in resected BM/plasma
Afatinib alone vs. afatinib + vinorelbine vs. investigator’s choice	Cortes et al.	LUX-Breast 3 (NCT01441596)	HER2-positive BCBM with recurrence or progression during or after trastuzumab and lapatinib	II	Patient benefit at 12 weeks
Tucatinib	Murthy et al.	HER2CLIMB (NCT02614794)	HER2-positive MBC previously treated with trastuzumab, pertuzumab, and Trastuzumab emtansine	II	PFS
Tucatinib + T-DM1	Borges et al.	NCT01983501	HER2-positive MBC	I	Incidence of adverse effects
Tucatinib + T-DM1 vs. T-DM1	Hurvitz et al.	HER2CLIMB-02 (NCT03975647)	HER2-positive MBC previously treated with a taxane and/or trastuzumab	III	PFS
Pyrotinib + vinorelbine	(Ongoing)	NCT03933982	HER2-positive BCBM	II	ORR of CNS
Pyrotinib + capecitabine	Yan et al.	PERMEATE (NCT03691051)	HER2-positive BCBM	II	ORR of CNS
GRN1005 (paclitaxel trevatide) vs. HRN1005 + trastuzumab	Bates et al.	GRABM-B	HER2-positive BCBM	II	ORR in CNS
GDC-0084 + trastuzumab	(Ongoing)	NCT03765983	HER2-positive BCBM	II	ORR in CNS
Trastuzumab Deruxtecan	Jerusalem et al.	DESTINY-Breast01 (NCT03248492)	HER2-positive MBC that had received previous treatment with T-DM1	II	ORR
Trastuzumab Deruxtecan vs. Trastuzumab emtansine	Cortés et al.	DESTINY-Breast03	HER2-positive unresectable and/or metastatic breast cancer previously treated with Trastuzumab and Taxane	III	PFS
Trastuzumab Deruxtecan vs. physician’s choice of treatment	Modi et al.	DESTINY-Breast04	Previously treated HER2-low advanced breast cancer	III	PFS
Etirinotecan pegol vs. physician’s treatment of choice	Cortes et al.	BEACON (NCT01492101)	Primary brain tumor or BM	III	OS
Erinotecan pegol vs. phisician’s treatment of choice	Tripathy et al.	ATTAIN	Stable BCBM previously treated with an anthracycline, a taxane, and capecitabine	III	OS
Everolimus + vinorelbine + trastuzumab	Van Swearingeng et al.	NCT01305941	HER2-positive BCBM	II	ORR in CNS
Alpelisib + capecitabine vs. buparlisib + capecitabine vs. buparlisib + capecitabine + trastuzumab vs. buparlisib + capecitabine + lapatinib	McRee et al.	NCT01300962	MBC, including BCBM	I	MTD and DLT
Buparlisib + fulvestrant vs. fulvestrant alone	Basega et al.	BELLE-2 (NCT01610284)	HR-positive/HER2-negative locally advanced or MBS with postmenopausal status	III	PFS
Buparlisib + fulvestrant vs. fulvestrant alone	Di Leo et al.	BELLE-3 (NCT01633060)	HER2-negative, locally advanced, or MBC that had relapsed on or after endocrine therapy and mTOR inhibitors	III	PFS
Buparlisib + capecitabine (+ trastuzumab in HER2-positive group)	Completed	NCT02000882	BCBM	II	CBR
Palbociclib	(Ongoing)	NCT02896335	Measurable progressive luminal-type BCBM	II	Clinical benefit rate at 8 weeks
Abemaciclib	Tolaney et al.	NCT02308020	BM from luminal-type BC, NSCLC, or melanoma	II	ORR in CNS
Abemaciclib or PI3K inhibitor GDC-0084 or entrectinib, selected by genetic test	(Ongoing)	NCT03994796	New or progressive BM	II	ORR in CNS
Palbociclib + trastuzumab + lapatinib + fulvestrant	(Ongoing)	NCT04334330	ER-positive/HER2-positive BCBM	II	ORR
Ribociclib + Letrozole	Hortobagyi et al.	MONALEESA-2 (NCT01958021)	Postmenopausal women with HR+, HER2-negative ABC	III	PFS
Ribociclib + Endocrine therapy vs. placebo + endocrine therapy	Tripathy et al.	MONALEESA-7 (NCT02278120)	Advanced HR+, HER2-negative BC	III	OS
Ribociclib + buparlisib + fulvestrant vs. ribociclib + alpelisib + fulvestrant vs. ribociclib + fulvestrant	Tolaney et al.	NCT02088684	Postmenopausal women with HR-positive/HER2-negative locally recurrent or advanced MBS	I/II	Incidence of DLTs and PFS
Ribociclib + letrozole	De Laurentis et al.	ComPLEEment-1	Advanced HR+, HER2- BC	IIIb	Safety, tolerability, and efficacy
Pembrolizumab	Nanda et al.	KEYNOTE-012 (NCT01848834)	Advanced TNBC, advanced head and neck cancer, advanced urothelial cancer, or advanced gastric cancer	I	Adverse events and overall response rate
Pembrolizumab + SRS	(Ongoing)	NCT03449238	BCBM	I/II	Tumor response at 8 weeks
Pembrolizumab vs. chemotherapy	Winer et al.	KEYNOTE-119	Metastatic TNBC	III	OS
Pembrolizumab + paclitaxel vs. pembrolizumab + nab-paclitaxel vs. pembrolizumab +gemcitabine/carboplatin vs. pembrolizumab + chemotherapy vs. chemotherapy	Cortes et al.	KEYNOTE-355	Previously untreated locally recurrent inoperable BC or metastatic TNBC	III	Adverse events and PFS
Atezolizumab + nab-paclitaxel vs. nab-paclitaxel	Schmid et al.	IMpassion130 (NCT01004172)	Unresectable locally advanced or metastatic TNBC	III	PFS and OS
Atezolizumab + paclitaxel vs. Paclitaxel alone	Miles et al.	IMpassion131 (NCT03125902)	Previously untreated locally advanced or metastatic TNBC	III	PFS
Atezolizumab + SRS	Active, not recruiting	NCT03483012	TN-type BCBM	II	PFS
Atezolizumab + chemotherapy vs. chemotherapy	Kyte et al.	ALICE (NCT03164993)	Locally advanced or metastatic TNBC	II	Toxicity and PFS
Carboplatin and bevacizumab	Leone et al.	NCT01004172	New or progressive BCBM	II	ORR in CNS
Bevacizumab, etoposide, and cisplatin	Wu et al.	NCT01281696	BC brain and or leptomeningeal metastases	II	ORR in CNS
Nivolumab + SRS	Ahmed et al.	NCT03807762	BCBM	II	Number of participants who experience DLTs
Talazoparib vs. single-agent chemotherapy of investigator’s choice	Litton et al.	EMBRACA (NCT01945775)	Advanced and/or MBC patients with BRCA mutation who received no more than three prior chemotherapy-inclusive regimens for locally advanced and/or metastatic disease	III	PFS
Olaparib vs. single-agent chemotherapy of investigator’s choice	Robson et al.	OlympiAD (NCT02000622)	MBC patients who had received no more than two previous chemotherapy regimens for metastatic disease	III	PFS
Veliparib + carboplatin + paclitaxel vs. Carboplatin + paclitaxel	Dieras et al.	BROCADE3 (NCT02163694)	Advanced HER2-negative BC with BRCA1 or BRCA2 mutation	III	PFS
Cisplatin + veliparib vs. Cisplatin alone	Sharma et al.	SOWG S1416 (NCT02595905)	Recurrent or metastatic TNBC, including BM	II	PFS
Rucaparib	Patsouris et al.	RUBY (NCT0250548)	HER2-negative MBC with BRCAness	II	CBR
Niraparib + pembrolizumab	Vinayak et al.	TOPACIO/KEYNOTE-162 (NCT02657889)	Advanced or metastatic TNBC or recurrent ovarian cancer	I/II	Number of subjects reporting DLTs and ORR
Sorafenib + WBRT	Morikawa et al.	NCT01724606	BCBM	I	MTD and toxicity by number of adverse events
HER2-CAR T cells	Recruiting	NCT03696030	HER2-positive recurrent BCBM or leptomeningeal metastases	I	Incidence of DLTs and number of adverse events
HER2-CAR T cells	(Ongoing)	NCT02442297	HER2-positive CNS tumors	I	ORR in CNS

Abbreviations: BC, breast cancer; MBC, metastatic breast cancer; BM, brain metastases; BCBM, breast cancer brain metastases; TN, triple-negative; TNB, triple-negative breast cancer; ABC, advanced breast cancer; HR, hormone receptor; RT, radiotherapy; WBRT, whole-brain radiotherapy; SRS, stereotactic radiosurgery; HER2, human epidermal growth factor receptor 2; OS, overall survival; CAR, chimeric antigen receptors; CBR, clinical benefit rate; CR, clinical response; PFS, progression-free survival; ORR, objective response rate; PS, performance status; CNS, central nervous system; T-DM1, trastuzumab emtansine; NSCLS, non-small-cell lung cancer; BRCA, breast-cancer gene; MRI, magnetic resonance imaging; MTD, maximum tolerated dose; DLTs, dose-limiting toxicities.

## Data Availability

Not applicable.

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
