# Peer review of "Advances in the Management of Central Nervous System Metastases from Breast Cancer"

_ijms, 2022, doi:10.3390/ijms232012525_

Round 1

Reviewer 1 Report

Jorge Avila and José Pablo Leone have shown in their review article the different CNS metastases from BC. The review article is sounds and collecting information from literature as one source for oncology readers. Only few references have to be added in the following lines; 

1. Line 39-41

2. Line 44-46

3. Abbreviations of WBRT and SRS at line 71 with references

Author Response

Thank you very much for your suggestions

1. Line 39-41

Reference was added "Miyazawa et al."

2. Line 44-46

Reference is added in line 47 "Sperduto et al."

3. Abbreviations of WBRT and SRS at line 71 with references

Abbreviations were presented on line 55 for WBRT and 56 for SRS respectively. Line 71 includes reference for "Kim et al."

Reviewer 2 Report

The manuscript entitled “Advances in the Management of Central Nervous System Metastases from Breast Cancer” by Jorge Avila and José Pablo Leone is well organized. The authors have done an excellent job putting together the information in the review article. The citing of the clinical trials is much appreciated.

Major comment:

The authors described WBRT and SRS but failed to mention SRT. I would suggest that authors include SRT as well in this review. Following are two articles as an example related to SRT in BCBM.

Ippolito E, Silipigni S, Matteucci P, Greco C, Carrafiello S, Palumbo V, Tacconi C, Talocco C, Fiore M, D'Angelillo RM, Ramella S. Radiotherapy for HER 2 Positive Brain Metastases: Urgent Need for a Paradigm Shift. Cancers (Basel). 2022 Mar 15;14(6):1514. doi: 10.3390/cancers14061514. PMID: 35326665; PMCID: PMC8946529.

Mampre D, Mehkri Y, Rajkumar S, Sriram S, Hernandez J, Lucke-Wold B, Chandra V. Treatment of breast cancer brain metastases: radiotherapy and emerging preclinical approaches. Diagn Ther. 2022;1(1):25-38. doi: 10.55976/dt.1202216523-36. Epub 2022 Jun 20. PMID: 35782783; PMCID: PMC9249118.

Author Response

Thank you for your suggestions

The revised manuscript includes this paragraph:

As mentioned before, fSRT has also been evaluated for BCBM. fSRT is less costly and more comfortable for patients [33] and retrospective reviews comparing fSRT to SRS have shown no difference in local control of BCBM [34], meanwhile, emerging studies suggest improved local control in fSRT when compared to SRS for BM [35, 36]. However, there is limited information on fSRT when compared to WBRT . The use of fSRT in 1-10 BM is currently being evaluated in the NCT04061408 study [37].

This includes the references the reviewer suggested as well

Line 71 includes SRT as part of the options for BCBM local treatment

Author Response

Thank you for your suggestions. 

We are addressing them in the following ways:

I am surprised not to see once the word blood brain barrier (BBB), which is the only issue in the treatment of brain metastases. Some physiological explanations and details about BBB should be given

We appreciate this suggestion, and we have included the following information about the BBB on the manuscript

The blood brain barrier (BBB), a neurovascular unit composed of endothelial cells, astrocytes and pericytes, makes treatment for BM challenging, as it forms a selective barrier between the CNS and the systemic circulation. The BBB gets disrupted during tumor progression and forms the blood tumor barrier (BTB), which is more permeable than the BBB, allowing multiple drugs into the CNS to target the tumor, and healthy brain parenchyma. The permeability of BTB varies depending on the subtype of cancer, for example BM from TNBC or basal type BC may often disrupt the BBB, whereas BM from HER2-positive BC tend to preserve the BBB.

Although there is no doubt the BTB is more permeable than the BBB, it still significantly restricts the delivery of anticancer drugs and obstructs systemic chemotherapeutics of brain tumors. RT, besides providing cytotoxicity, can cause disruption of the BBB resulting in an increased permeability into the surrounding brain parenchyma. A better understanding of both the BBB and BTB is needed to develop treatments with higher penetration to the CNS or increased manipulation of the CNS barrier.

At the beginning of paragraph 3 there should be a sentence explaining the difficulties associated with the BBB.  A problem of diffusion through an impermeable barrier, especially for large molecules such as therapeutic antibodies or chemotherapy. A problem of rapid efflux of drugs to the outside of the CNS, which does not allow them to remain in effective pharmacological concentrations to treat BCBMs

We thank the reviewer for this suggestion and have included the following information on the manuscript

Plasma proteins such as immunoglobulin G (IgG) molecules generally do not cross the brain capillary endothelial wall, which forms the BBB under normal conditions. However, certain monoclonal antibodies in the circulation do cross the BBB by a process of receptor mediated transcytosis. In this case, the monoclonal antibody is directed against a specific receptor located on the luminal membrane of the brain capillary endothelium, and the monoclonal antibody binds to exofacial epitopes on the BBB receptors. The reverse transcytosis of IgG molecules across the BBB is most likely mediated by an Fc receptor situated on the abluminal membrane of the brain capillary endothelium, and rat models have demonstrated a rapid IgG efflux with a halftime of 48min. In the following paragraphs we will mention different drugs that have been evaluated for BCBM, however, their passage through the BBB/BTB is still under study.

Paragraph 3.1.1: it must be explained that the problem of therapeutic antibodies is their weak passage through the BBB (widely found in the literature), but especially their rapid efflux to the blood, through the FcRn receptor (there is no evidence but this hypothesis is widely accepted, see article by Zhang and Pardridge, 2001).

This was addressed in the previous paragraph (mentioned above), including the suggested reference. However, we also added the following sentences

Under impaired BBB conditions such as meningeal carcinomatosis or RT, trastuzumab levels in cerebrospinal fluid (CSF) are increased; this evidence support the concept of continuing trastuzumab therapy in patients with BM treated by RT. Monitoring of trastuzumab in the serum and CSF may enable individualized therapies in BCBM”

“Paragraph 3.1.7: Tucatinib has been specifically developed to circumvent efflux phenomena by PgP, which is why it is particularly effective on BCBMs.”

The first part of the paragraph 3.1.7 was modified to mention these elements:

“Tucatinib is a specific inhibitor of HER2 and a substrate of P glycoprotein (P-gp) which would expect to limit distribution to CNS, however, reduced number of efflux transporters, acid interstitial pH and leaky junctions enhance tucatinib permeability into the tumor.”        

“We could have some examples of molecules in clinical trials: The case of Angiopep (especially ANG1005), a molecule being tested for BCBMs, which uses a combination of chemotherapy with a BBB lipoprotein receptor recognition domain. Thus, the molecular can penetrate like a "Trojan horse" into the CNS. This molecule has been analyzed in phase II with promising results (Kumthecar, Clin Cancer Res, 2020)”

We included this and other references about ANG1005 in the article

ANG1005, a novel taxane derivative consisting on 3 paclitaxel molecules covalently linked to Angiopep-2, designed to cross the BBB and BTB and to penetrate malignant cells via the lipoprotein receptor related protein 1 (LRP-1) transport system has demonstrated intracranial response, symptom improvement and prolonged overall survival compared to historical controls in different studies on BCBM. Unlike native paclitaxel, ANG1005 was proven not to be a substrate for the multiple drug resistance (MDR) efflux pump in in vitro studies and has shown similar brain penetration in mice. This indicates that ANG1005 can be used to treat both peripheral metastatic BC and BCBM even if the tumor develops resistance to conventional taxanes.

In paragraph 3.1.1, other approaches such as intrathecal injection used for a very long time, but for example a clinical case of repeated intrathecal injections of tratuzumab to circumvent the phenomenon of antibody efflux, having allowed prolonged control of BCBMs (Bousquet, JCO, 2016).

We agree with this suggestion, and we have added the following paragraph

Several case reports have described a benefit of intrathecal trastuzumab administration to treat carcinomatous meningitis in patients with HER2-overexpressing metastatic BC. Bousquet et al. described the effect of intrathecal trastuzumab in a patient with HER-2 BCBM; and after maintaining an efficacious concentration of this drug in the CSF, they were able to achieve stabilization of brain and epidural metastases previously resistant to radiation and chemotherapy   More recently Kumthekar et al. evaluated intrathecal trastuzumab in a multicenter phase I/II study on patients with HER2-positive leptomeningeal disease (LMD). Partial response was observed in 19.2% of the patients and stable disease was seen in 50% of the participants. This study suggests promise for potentially improved outcomes of HER2-positive LMD. However, further analysis needs to be done for the evaluation of intrathecal trastuzumab.

Round 2

Reviewer 3 Report

The authors have taken into account all of remarks. It is a complete work that can in my opinion be published

Author Response

We thank the reviewer for their comment. They mention the article has already taken into consideration all the remarks they suggested and can be published.